# Availability and utilization of sexual and reproductive health services among adolescents of Godawari Municipality, Nepal: A cross-sectional study

Laxmi Gautam [ID][1*], Aastha Maharjan [ID][1], Harikrishna Bhattarai[1], Sujan Gautam [ID][1,2]

**1** Department of Public Health, Manmohan Memorial Institute of Health Sciences, Tribhuvan University, Kathmandu, Nepal, **2** Institute of Fundamental Research and Studies (InFeRS), Kathmandu, Nepal

\* laxmi.dp26@gmail.com, 45laxmigautam@gmail.com

## Abstract

Adolescent sexual and reproductive health is getting due attention in developing countries because investing among adolescents can bring multiple benefits. This study aims to assess the availability and utilization of sexual and reproductive services (SRHS), and explore different factors affecting the utilization of SRHS by adolescents of Godawari Municipality. A cross-sectional, mixed-method study was conducted using Anderson's model of health service utilization in three government secondary schools and three local health institutions in Godawari Municipality. A self-administered questionnaire was used with adolescents aged 15–19 (n = 416), and conducted Key Informants Interview (KII) and observation were conducted in health institutions. Quantitative data was analyzed in Statistical Package for Social Sciences (SPSS) version 16, and thematic analysis was done for qualitative data. The mean age of the respondents was 17.31 ± 0.99 years. Nearly one-third of them (30.63%) had not heard about Adolescent Friendly Health Services. Knowledge of ASRH and AFHS (AOR = 2.814, 95% C.I = 1.129-7.013) and conversation with parents (AOR = 2.069, 95% C.I = 1.094-3.912) and availability of ASRH (AOR = 2.801, 95% C.I = 0.810-9.687) had a significant relationship with utilization of health services. The adolescents' perceived feeling of the need for ASRH services is significantly associated with the utilization of ASRH services (p < .001). Only 13.22% of them had ever used ASRH services, and the reasons for not using services were the lack of realization of the need for services (60.39%), while 14.55% of them felt that privacy was not maintained at all. The KII found the number of adolescents visiting Health Facility (HF) was very low, and those visiting for SRH-related services were rare. The observation showed none of the HF met the adolescent friendly criteria set by the government even when they were once certified. Knowledge of respondents and their parents, as well as availability of the services, had a significant role in the utilization

**Data availability statement:** All relevant data generated in the study are included in the paper and its Supporting Information files.

**Funding:** The authors received no specific funding for this work.

**Competing interests:** The authors have declared that no competing interests exist.

of ASRH services by the adolescents. Awareness about the importance of health care services among adolescents and the availability of services is essential.

## Introduction

The World Health Organization (WHO) definition officially designates an adolescent as someone between the ages of 10 and 19. Adolescents experience rapid physical, cognitive, and psychosocial growth. Their emotions, thoughts, decisions, and interactions are affected by this [1]. Adolescent Sexual and Reproductive Health (ASRH) refers to the physical and emotional wellbeing of adolescents and includes their ability to remain free from unwanted pregnancy, unsafe abortion, sexually transmitted infections (including HIV/AIDS), and all forms of sexual violence and coercion [2]. The health needs of young people are special, so they need special services called Adolescents Friendly Health Services (AFHS), which are defined by the friendly environment and favorable condition where adolescents get access to ASRH services in an accessible, equitable, acceptable, appropriate, and effective manner [3]. However, little information exists regarding what Nepalese adolescents perceive as their SRH needs, their service utilization patterns, and factors that act as barriers or facilitate the use of existing services [4].

Adolescents in Nepal do not have adequate access to information and access to services, which is further exacerbated by the little sex education in schools and hardly open discussion about sex and sexuality in families and society [5]. There are limited sessions on adolescents' sexual and reproductive health in the school curriculum. Therefore, adolescents are unable to receive complete information about their physiological, psychological, and emotional changes that occur during puberty [6]. To meet the health needs of adolescents, the government of Nepal launched a national program in 2010 to provide adolescent-friendly sexual and reproductive health services as part of its five-year health sector plans [7]. Nepal Demographic Health Survey (NDHS) 2022 revealed that 14% of women aged 15–19 have ever been pregnant, 10% have had a live birth, and 2% have had a pregnancy loss. Two percent each of women and men aged 15–19 had sexual intercourse before age 15. By age 15, three percent of women aged 15–19 were married, and 1% had already become pregnant [8]. Although SRH education and services are provided to young people, unplanned pregnancy, unsafe abortion, and sexually transmitted infections commonly exist among adolescents in Nepal, indicating underutilization of sexual health services by young people. Health facilities have apparently failed to provide young people with specialized sexual health education and services [9]. However, despite the interest of young people in obtaining relevant information and friendly services, the provision of sexual and reproductive health services in Nepal is very inadequate, and little information exists at the national level regarding what Nepalese adolescents perceive as their SRH needs, their service utilization patterns, and factors that act as barriers or facilitates use of existing services [10].

Global evidence shows that programs such as comprehensive sexuality education and youth-friendly services have a positive influence on SRH service utilization [11]. Poor sexual and reproductive health knowledge, the lack of youth-friendly services, the lack of confidentiality of services, experiences of shame, and health care provider attitudes also have an influence on SRH service utilization by adolescents [12]. Correct evidence is crucial to properly planning SRH services for Nepalese adolescents. Amongst the many means to measure adolescents' needs, their own reports of health behaviors and lifestyles and utilization of health services are especially valuable. So, this study aims to assess the availability and utilization of ASRH services and explores different factors affecting the utilization of the services related to individuals, health care institutions, and service providers.

## Materials and methods

A cross-sectional study was conducted to assess the availability and utilization of ASRH services by using quantitative and qualitative methods from November 2021 to January 2022. The dependent variable was the utilization of SRH services by adolescents, and the independent variables were socio-demographic factors and health service-related factors. The study was conducted in Godawari Municipality, Lalitpur District, Nepal. Godawari Municipality was chosen purposively because it represents semi-urban areas and was formed by merging the former 12 Village Development Committees (VDCs). The study is delimited to the government schools to illustrate the situation of the semi-urban population in a resource-limited setting. Private school students perform better than public school students, mostly in Nepal. If both public and private schools were selected, there could be a positive private school effect in the findings of this study [13]. In addition, the public schools lack resources to get access to information and opportunities, which may differ from the situation for the students of private schools, who may have better opportunities to access information and health care services [14].

For the quantitative study, the list of government higher secondary schools in Godawari Municipality was obtained from the municipal office. Of the total ten government schools, three schools were selected randomly using the lottery method. A total of 416 school-going adolescents aged 15 to 19 years studying in grades 11th and 12th of those were selected. The sample size was calculated using Cochran's formula for an infinite population [15], taking the reference prevalence rate of SRH utilization as 44% [16], a 95% confidence level, a 5% allowable error, and a non-response rate of 10%. So the total 416 samples included the 378 actual sample sizes and added 10% of the sample size. The number of students in each school and class was selected using probability proportionate sampling (PPS). Each individual participant was selected randomly using the lottery method.

For the qualitative study, three health service providers in the local-level health institutions certified as adolescent-friendly in close proximity to those selected schools were chosen purposively. The local health institutions were studied to figure out the providers' perspectives only; however, the utilization of ASRHS by the study participants was not limited to local health institutions. They reported the services they used from anywhere.

During data collection, permission was taken from the municipality health section and education section, and information about schools and health institutions was gathered. The information about the adolescent-friendly certified health institutions of the municipality was collected from the family welfare division. Then, schools were contacted, and permission was obtained from the school administration before collecting data from the students.

Quantitative data was collected through a self-administered, semi-structured questionnaire, and the qualitative data through in-person interviews guided by an interview guideline. The questionnaire was developed using Anderson's model of health service utilization [17–18].The predisposing, enabling, need, and external environmental factors were included in the questionnaire [19–20]. These factors were analyzed separately during data analysis. The health institutions were observed using a checklist. The data collection tools were developed after an extensive literature review based on various previously conducted studies and were pretested in Kageswori Manohara Municipality of the Kathmandu district. The observation checklist was adapted from the Government of Nepal's criteria for Adolescent Friendly Health Institutions.

Pretesting was done among 10% of the sample size and in one health institution to check instruments, and a few modifications were made. The questionnaire was translated into the Nepali language.

For the analysis of quantitative data, IBM-SPSS software version 16 was used. Descriptive statistics were used to study the characteristics of variables. The Chi-square test was used to test the association, and logistic regression modeling was done to predict the utilization of ASRH services.

The qualitative information from the interviews was noted and recorded with the permission of the interviewee. They were transcribed into Nepali and then translated into English, which was analyzed through the process of thematic analysis, identified primary themes in the information and categorized them into similar themes. The crucial details from the observation checklist were noted.

### Ethical statement

The proposal was approved by the Institutional Review Committee (IRC) at the Manmohan Memorial Institute of Health Science (MMIHS-IRC 619). The study was conducted from 26th November 2021 to 17th January 2022. Written informed consent was taken with each respondent before data collection for the quantitative study, while verbal informed consent was taken for the qualitative study. In the case of participants below the age of 18, permission was taken from the school principal and their respective class teachers, and then assent was taken from each of them. Participation was voluntary, and all ethical considerations, including confidentiality and privacy, were maintained throughout the study process.

### Result

The mean age of the respondents was $17.31 \pm 0.99$ years. More than half of the participants were 15–17 years old (57.87%), female (55.05%), and were studying in class 12 (55.67%). Only 13.83% of the participants were married or in a relationship, and the relationship status was statistically associated ($p = 0.024$) with utilization of ASRH. Only about one-third of the respondents (30.53%) had conversations with their parents about various aspects of ASRH. However, factors such as age, sex, and the class they study in were not significantly associated with utilization. Regarding the educational status of parents, 13.89% of them had an illiterate father, which was 28.15% in the case of the mother.

Almost all (99.28%) of the respondents knew about or were aware of ASRH. Nearly one-third of them (30.63%) had not heard about Adolescent Friendly Health Services. More than three-quarters (77.40%) of them knew about any health institutions that provided ASRH services in their area. Knowledge of ASRH, AFHS, and conversation with parents had a significant relationship with utilization of Adolescent Sexual and Reproductive Health Services (ASRHS). (Table 1)

Regarding the need factors of ASRHS utilization, almost half of the respondents reported that they had felt the need for ASRH services (48.56%). The adolescents' perceived feeling of the need for ASRH services is significantly associated with the utilization of ASRH services ($p < 0.001$). The majority of the respondents (88.43%) felt the need for separate and specific services for adolescents. Nearly two-thirds (65.62%) thought that their decision on ASRH service utilization would be affected by the gender of the service provider. However, their perception of the need for separate and specific services for adolescents as well as the effect of the sex of the service provider were not significantly associated with the utilization of ASRH services. Only two-thirds (66.59%) of the respondents thought they would use the service in the future if needed. The willingness to utilize the ASRH service in the future was significantly associated with the services utilization.

In the case of external environmental factors, for more than 2/3rd of respondents (68.94%), the nearest health institution was less than 30 minutes walking distance from their home. However, information and awareness about ASRH and the distance to the health institution from home were not significantly associated with the service utilization. The majority of the respondents (82.13%) reported the need for a convenient opening hour, which was significantly associated with the utilization of ASRHS. (Table 2)

The respondents who had conversation with their parents about ASRH were 2.069 times more likely to utilize ASRH services than the respondents who did not have conversation with parents on ASRH-related matters (AOR = 2.069, 95%

PLOS Global Public Health

**Table 1. Predisposing and enabling factors of ASRHS utilization.**

| Variables | Utilization of ASRHS | | Total | | P value |
|---|---|---|---|---|---|
| | Yes (%) | No (%) | Frequency | % | |
| **Predisposing factors** | | | | | |
| Age†: Mean ±SD = 17.31 ± 0.99 years | | | | | |
| ≤ 17 years | 30 (12.60) | 209 (87.40) | 239 | 57.87 | 0.592 |
| > 17 years | 25 (14.40) | 149 (85.60) | 174 | 42.13 | |
| **Sex** | | | | | |
| Male | 26 (13.90) | 161 (86.10) | 187 | 44.95 | 0.710 |
| Female | 29 (12.70) | 200 (87.30) | 229 | 55.05 | |
| Level of Education | | | | | |
| Grade 11 | 24 (12.70) | 165 (87.30) | 189 | 45.43 | 0.774 |
| Grade 12 | 31 (13.70) | 196 (86.30) | 227 | 55.67 | |
| Religion | | | | | |
| Hindu | 39 (12.80) | 266 (87.20) | 305 | 73.32 | 0.665 |
| Others | 16 (14.40) | 95 (85.60) | 111 | 26.68 | |
| Relationship status† | | | | | |
| Married/ In Relationship | 13 (22.80) | 44 (77.20) | 57 | 13.83 | 0.024 |
| Single/ Others | 42 (11.80) | 313 (88.20) | 355 | 86.27 | |
| **Enabling Factors** | | | | | |
| Father's Education† | | | | | |
| Illiterate | 8 (14.30) | 48 (85.70) | 56 | 13.89 | |
| Primary school | 12 (10.30) | 105 (89.70) | 117 | 29.03 | 0.756 |
| Secondary school | 22 (15.30) | 122 (84.70) | 144 | 35.73 | |
| High school | 6 (16.70) | 30 (83.30) | 36 | 8.93 | |
| Literate only | 6 (12.00) | 44 (88.00) | 50 | 12.40 | |
| Mother's Education† | | | | | |
| Illiterate | 15 (87.10) | 101 (12.90) | 116 | 28.15 | 0.900 |
| Primary school | 14 (14.10) | 85 (85.90) | 99 | 23.91 | |
| Secondary school | 10 (12.20) | 72 (87.80) | 82 | 1.98 | |
| High school | 1 (6.30) | 15 (93.80) | 16 | 3.86 | |
| Literate only | 15 (14.90) | 86 (85.10) | 101 | 24.40 | |
| Heard About ASRH | | | | | |
| Yes | 55 (13.30) | 358 (87.00) | 413 | 99.28 | 1.000++ |
| No | 0 (0.00) | 3 (100.00) | 3 | 0.72 | |
| Known the Institutions providing ASRH | | | | | |
| Yes | 52 (16.10) | 270 (83.90) | 322 | 77.40 | 0.004++ |
| No | 0 (0.00) | 25 (100.00) | 25 | 6.00 | |
| Don't Know | 3 (4.30) | 66 (95.70) | 69 | 16.60 | |
| Heard about AFHS | | | | | |
| Yes | 49 (16.90) | 240 (83.10) | 289 | 69.47 | 0.001 |
| No | 6 (4.70) | 121 (95.30) | 127 | 30.63 | |
| Conversation with parents about ASRH | | | | | |
| Yes | 26 (20.50) | 101 (79.50) | 127 | 30.53 | 0.004 |
| No | 29 (10.00) | 260 (90.00) | 289 | 69.47 | |
| Parents support in use of services † | | | | | |
| Yes | 46 (14.60) | 270 (85.40) | 316 | 76.51 | 0.181 |
| No | 9 (9.30) | 88 (90.70) | 97 | 23.49 | |

Note: †Missing data, therefore total sample may not be equal to 416, ++Fisher's exact test.

**Table 2. Need and External Environmental factors of ASRHS utilization.**

| Variables | Utilization of SRHS | | Total | | P value |
|---|---|---|---|---|---|
| | Yes (%) | No (%) | | | |
| **Need Factors** | | | | | |
| Felt need of service | | | | | |
| Yes | 55 (27.22) | 147 (72.78) | 202 | 48.56 | <.001 |
| No | 0 (0.00) | 214 (100) | 214 | 51.44 | |
| Perceived need of separate service † | | | | | |
| Yes | 48 (13.10) | 319 (86.90) | 367 | 88.43 | 0.773 |
| No | 7 (14.60) | 41 (85.40) | 48 | 1157 | |
| Use in future if needed | | | | | |
| Yes | 49 (17.80) | 227 (82.20) | 276 | 66.35 | 0.001 |
| No | 2 (7.40) | 25 (92.60) | 27 | 6.49 | |
| Don't Know | 4 (3.50) | 109 (96.50) | 113 | 27.16 | |
| **External environmental factors** | | | | | |
| Distance to health institution † | | | | | |
| <30 minutes walking distance | 34 (15.30) | 188 (84.70) | 222 | 68.94 | 0.641 |
| 30 minutes to 1 hour walking distance | 11 (16.20) | 57 (83.80) | 68 | 21.12 | |
| >1 hour walking distance | 7 (21.90) | 25 (78.10) | 32 | 9.94 | |
| Necessity of Convenient Opening hours of institution | | | | | |
| Yes | 53 (15.60) | 287 (84.40) | 340 | 82.13 | 0.010 |
| No | 2 (2.70) | 72 (97.30) | 74 | 17.87 | |
| Perceived Community support | | | | | |
| Yes | 38 (15.70) | 204 (84.30) | 242 | 58.31 | 0.082 |
| No | 17 (9.80) | 156 (90.20) | 173 | 41.69 | |
| Perceived Religious and cultural beliefs/Barriers | | | | | |
| Yes | 20 (14.40) | 119 (85.60) | 139 | 33.41 | 0.619 |
| No | 35 (12.60) | 242 (87.40) | 277 | 66.59 | |
| Perceived Effect of sex of service provider † | | | | | |
| Yes | 36 (13.30) | 235 (86.70) | 271 | 65.62 | 0.862 |
| No | 18 (12.70) | 124 (87.30) | 142 | 34.38 | |

Note: †Missing data, therefore total sample may not be equal to 416.

C.I = 1.094–3.912). The respondents who knew about AFHS were 2.8 times more likely to utilize services than the respondents who were not aware of AFHS (AOR = 2.814, 95% C.I = 1.129–7.013). The respondents who knew about the health institutions providing ASRH services are 4.237 times more likely to use services compared to the ones who had no idea whether these services are available or not (AOR = 2.801, 95% C.I = 0.810 – 9.687). Because there were no respondents who used ASRHS without the felt need of services and knowledge about institutions providing ASRH, logistic regression was not applicable even though they were significant factors in utilizing ASRHSs. (Table 3)

## Characteristics related to utilization of services by respondents

The respondents made use of ASRH services. Very few of them (13.22%) had ever used ASRH services; among them, the majority used those services in government health institutions (69.01%). Most of them got general health checkups and general health information counseling services, and condom collection and family planning services. The reasons for not using services were mostly because they did not think they needed the service (60.39%). Other reasons were

them not being interested in using such services (38.23%), being ashamed to ask for services (26.31%), and not knowing where to get such services (20.22%). Some of them (7.48%) also responded that the opposite sex of the service providers was also the reason. Around half of them (52.73%) had paid for the services. Most of them (81.82%) had seen IEC materials in the institute. Around 40.00% did not have to wait or only waited <30 minutes for services. Three-fourths (70.91%) of them were comfortable asking the questions. nearly all (90.91%) expressed satisfaction with the attention they receives from health workers. Less than one fourth (14.55%) expressed dissatisfaction with the lack of privacy protection. Almost all of them (94.55%) would recommend others go to the health institutions. (Table 4)

The key informant interview (KII) among health workers revealed a low number of adolescents visiting health institutions, particularly those seeking SRH-related services. However, there was no separate recording system for adolescents, so accurate information about the number of adolescent visits was unavailable. The ones visiting the institutions would come for general reasons like fever, first aid, or other OPD services. The barriers identified in the delivery of services to adolescents were that the infrastructure was not adolescent-friendly, such as separate rooms, proper waiting spaces, the unavailability of appropriate IEC materials, and the unavailability of trained health workers to provide specific ASRH services. The health institutions only provided limited services like general health check-ups, health information, counseling services, family planning services, and pregnancy tests. They were not able to give comprehensive services to adolescents.

While observing the health institutions, all three were once certified as adolescent-friendly health institutions, but currently, they did not meet any specific criteria of being adolescent-friendly. There was no supply of ASRH-specific IEC materials, no trained health workers, no adolescent job aid was available, none of them had displayed the AFS logo, reporting in AFs format was null, and involvement of the adolescents in HFOMC was not in practice. Criteria met to some

**Table 3. Factors associated with utilization of ASRH services by using multiple logistic regression.**

| Factor | Utilization of services | | 95%CI (lower-upper limit) | COR | p-value | 95%CI (lower-upper limit) | AOR | p-value |
|---|---|---|---|---|---|---|---|---|
| | Yes n (%) | No n (%) | | | | | | |
| **Conversation with parents about ASRH** | | | | | | | | |
| Yes | 26 (20.50) | 101(79.50) | 1.296 – 4.110 | 2.308 | 0.005 | 1.094 – 3.912 | 2.069 | 0.25 |
| No | 29 (10.00) | 260 (90.00) | Ref | | | Ref | | |
| **Relationship status** | | | | | | | | |
| Married/ In relationship | 13 (22.80) | 44 (77.20) | 1.096 – 4.423 | 2.202 | 0.027 | 1.072 – 4.908 | 2.294 | 0.032 |
| Single | 42 (11.80) | 313 (88.20) | Ref | Ref | | | | |
| **Heard about AFHS** | | | | | | | | |
| Yes | 49 (16.90) | 240 (83.10) | 1.716 – 9.881 | 4.117 | 0.002 | 1.129 – 7.013 | 2.814 | 0.026 |
| No | 6 (4.70) | 121 (95.30) | Ref | | | Ref | | |
| **Knowledge about the availability of health institution providing ASRH services** | | | | | | | | |
| Yes | 52 (16.10) | 270 (83.90) | 1.283 – 13.990 | 4.237 | 0.018 | 0.810 – 9.687 | 2.801 | 0.104 |
| No | 0(0.00) | 25 (100.00) | 0.00 | 0.000 | 0.998 | | 0.00 | 0.998 |
| Don't Know | 3 (4.30) | 66 (95.70) | Ref | | | Ref | | |
| **Comfortable opening hours** | | | | | | | | |
| Yes | 53 (15.60) | 287 (84.40) | 1.583-27.872 | 6.648. | 0.010 | Ref | | |
| No | 2 (2.70) | 72 (97.30) | Ref | | | 0.039 – 0.716 | 0.167 | 0.16 |
| **Willingness to Use service in future** | | | | | | | | |
| Yes | 49 (17.80) | 227 (82.20) | 2.070–16.717 | 5.882 | 0.001 | 1.623 – 13.997 | 4.766 | 0.005 |
| No | 2 (7.40) | 25 (92.60) | 0.378–12.572 | 2.180 | 0.383 | 0.323 - 12.278 | 1.993 | 0.457 |
| Don't know | 4 (3.50) | 109 (96.50) | Ref | | | Ref | | |

Note: †Missing data, therefore total sample may not be equal to 416.

**Table 4. Characteristics related to experience during utilization of ASRH services.**

| Characteristics | Frequency | Percentage (%) |
|---|---|---|
| **Used ASRH** | | |
| Yes | 55 | 13.22 |
| No | 361 | 86.78 |
| Reason for not using service ⭱¶ (Among those who did not use ASRH services) | | |
| Don't know where to get services | 73 | 20.22 |
| Not interested in such service | 138 | 38.23 |
| Fear of parents | 22 | 6.09 |
| Behavior of service provider | 2 | 0.56 |
| Religion/Culture barrier | 12 | 3.32 |
| Ashamed to ask for services | 95 | 26.31 |
| Inconvenient opening hours | 22 | 6.09 |
| No Money to get services | 11 | 3.05 |
| Health facility's distance from home | 15 | 4.16 |
| opposite sex of service provider | 27 | 7.48 |
| Did not think service was needed | 218 | 60.39 |
| **Institution where service use⭱ (Among those who used ASRH services)** | | |
| Government institutes | 38 | 69.09 |
| Private institutes | 17 | 30.91 |
| **Service Used ⭱¶** | | |
| General Health Check up | 35 | 63.63 |
| Family planning | 6 | 10.91 |
| General health information | 33 | 60.00 |
| Condom collection | 11 | 20.00 |
| Counseling services | 13 | 23.64 |
| Others | 4 | 7.28 |
| **Paid for the service** | | |
| Yes | 29 | 52.73 |
| No | 26 | 47.27 |
| **Sex of service provider** | | |
| Male | 20 | 36.36 |
| Female | 35 | 63.64 |
| **IEC Materials in HF** | | |
| Yes | 45 | 81.82 |
| No | 10 | 18.18 |
| **Waiting time during service use** | | |
| No waiting | 22 | 40.00 |
| < 30 min | 24 | 43.63 |
| > 30 min | 9 | 16.37 |
| **Comfort to ask questions** | | |
| Yes | 39 | 70.91 |
| No | 16 | 29.09 |
| **Proper attention by HW** | | |
| Yes | 50 | 90.91 |
| No | 5 | 9.09 |

*(Continued)*

| Characteristics | Frequency | Percentage (%) |
|---|---|---|
| **Maintenance of privacy** | | |
| Not at all | 8 | 14.55 |
| maintained but not enough | 16 | 29.09 |
| Satisfactory | 10 | 18.18 |
| very good/ maintained well | 21 | 38.18 |
| **Recommend others** | | |
| Yes | 52 | 94.55 |
| No | 3 | 5.45 |

Note: † Missing data, therefore total sample may not be equal to 416,

¶ Multiple responses; therefore, the total percentage exceeds 100%.

extent were the convenient location of the health institution, toilets with clean water and dustbins, maintenance of privacy as far as possible while providing service, and counseling. The health institutions, once certified as AF institutions, were unable to maintain this status due to inadequate monitoring and follow-up. (Table 5)

## Discussion

This study indicates that respondents who were either married or in a relationship had a 2.202-fold higher likelihood of using ASRH services compared to those who were single (Crude OR=2.202, 95% C.I=1.096-4.423). Studies conducted in Bhaktapur and Surkhet among secondary school students (10–24 years) corroborate this, revealing that 77.08% of the married youths benefitted from the services more than their unmarried counterparts [21–22]. In Bhaktapur, researchers found that married individuals were six times more likely to utilize ASRH services compared to unmarried ones (Crude OR: 6.51) [21]. This may be due to the fact that there is stigma attached to the use of SRH-related services by young and unmarried individuals, but for those who are married or in a relationship, they are more sexually active, and their needs for SRH services increase, resulting in the use of such services.

This study identified that the respondents who had conversations with their parents about ASRH were 2.069 times more likely to utilize ASRH services than those who did not. Similar were the findings in the study done in Ethiopia, where adolescents were 10 times more likely to utilize VCT services among those who had a parental discussion compared to those who hadn't [23]. Moreover, parental communication is often regarded as the protective factor for adolescent health. They are potentially an important source of ASHR-related information to their children and can greatly influence children's attitudes and behaviors towards their health, including ARSH. Parental discussion promotes the likelihood of preventing the adaptation of unhealthy ARSH behaviors and practices, leading to the promotion of healthy lifestyles [24]. Students who have communicated SRH-related concerns with their parents play a significant role in promoting adolescent intentions for healthy ASRH behaviors and practices [25]. Another study conducted in India also revealed positive effects of parental communication in the utilization of ASRHS, mentioning that decreasing adolescent-parents communication related to sexual issues was associated with nonutilization of ASRH services [26]. Additionally, parental communication leads to reducing risky sexual behavior as well as utilizing adolescent-friendly services related to ARSH [24]. These findings from similar cultural contexts are supportive findings to depict the importance of parents and adolescents' communication in the issues related to sexual and reproductive health. However, it contradicts the study among youths in Kathmandu, where respondents having distinct interaction with parents were 2 times more likely to utilize SRH services than those having close interaction (COR: 2.424) [27]. The difference may have been due to the difference in the nature of

**Table 5. List of codes obtained on Thematic Analysis.**

**Theme: Utilization of SRH services by adolescents from the perspective of service providers**

| Domain | Codes | Codes description | |
|---|---|---|---|
| Current situation of ASRH service utilization | I | Very few adolescents visit health institutions, and even fewer for SRH-specific issues. | |
| | | They visit health institutions mainly for height/weight, blood pressure measurement, and to collect condoms (boys) and abdominal pain during menstruation (girls). | |
| | | Available services: general health check-ups, health information, counseling family planning devices, pregnancy tests | |
| Barriers for adolescents to utilize SRH services | II | Not well-informed about available services | Have privacy concerns |
| | | They are unable to identify their needs and seek assistance. | The service provider's gender. |
| | | No trust in the quality of services/competency of HWs of government health institutes. | |
| Barriers for health institutions to provide ASRH services | III | Infrastructure is not Adolescent Friendly | |
| | | There are trained health workers for ASRH service delivery | |
| | | It is not able to provide all the necessary services needed for adolescents | |
| | | No supply of IEC materials related specifically to adolescents | |
| | | Lack of coordination with stakeholders like municipality and schools | |
| | | No follow-up and monitoring of the adolescent friendly status of the institution | |
| Possible solutions (from health institutions) | IV | Reaching out with information about ASRH and services to adolescents/ awareness raising<br>Mobilize FCHV's in community<br>Discussions in mothers group meetings<br>Coordination with school<br>Through mothers who come for checkups<br>Distribute the IEC materials<br>Co-ordinate with schools to conduct School health programs and take health education classes for students.<br>Organize camps specific for adolescents<br>Provide service by same-sex health worker as far as possible<br>Maintain privacy and confidentiality. | |

participants in both studies, as the study in Kathmandu included participants up to 24 years old who may have been able to independently decide to utilize services and did not need communication with parents.

Around three-quarters (77.40%) of them knew about any health institutions that provided ASRH services in their area, and more than two-thirds of them had heard about AFHS. The respondents who were aware of these were more likely to use services compared to the others. It aligns with previous studies conducted in Bhaktapur and Dang, which also found a significant association [21,28]. This is because lack of information about the kind of ASRH services available is one of the important barriers to utilizing services, so having any awareness about ASRH or AFHS would be an enabling factor towards acquiring those services. For the many respondents (68.94%), the nearest health institution was less than 30 minutes walking distance from their home. Even though distance was not identified as a major barrier to utilization of services in this study, other studies in Bhaktapur and Dhading have identified it as an important aspect of accessing ASRH services [21,29].

Almost half of the respondents reported that they had felt the need for ASRH services in this study, which is similar to the study in Pyuthan (44%), but is very high compared to the study in Bhaktapur (15%) among participants within the last 12 months [21,30]. The figure is higher in this research as it collects the information of their needs in their entire life, whereas the one in Bhaktapur only takes into consideration the last 12 months from the time of data collection. Two-thirds

(66.59%) of the respondents thought they would use the service in the future if needed, which is almost double that of the study in Pokhara (34.70%) and Bhaktapur (37.00%) [21,31].

This study found that the overwhelming majority of adolescents were not utilizing SRH services. Only 13.22% of participants have ever used SRH services. The reasons for not utilizing ARSH services were mostly personal and health system-related factors. Some of the personal factors were: need was not perceived (60%); not interested in such services (38%); ashamed to ask for services (26%), etc. Likewise, health system-related factors include inconvenient opening hours (6%); access to health facilities (4%); health workforce-related (8%), finance-related (3%), etc. The qualitative study also confirms this finding. In the KII, the health workers reported that adolescents rarely utilize services, particularly those specific to SRH-related issues. Likewise, in the studies of similar nature conducted in Bhaktapur in 2015 and Pokhara in 2018, the utilization was only 9.2% and 17%, respectively [21,31]. Similarly, the household surveys in Kathmandu valley among the youths (15–24 years) and Bhaktapur district among adolescents (10–19 years) in 2020 also showed low utilization of services, which was 23% and 24.7% respectively [32,33]. A recent survey conducted in three states across India found that young men and women (aged 15–24 years) seldom consulted CHWs for issues related to sexual health [34]. The service use is lower in this study than that demonstrated in another study conducted in Pyuthan (44.0%) and also in Myanmar where around two-thirds of youths used some RH services at least once in the past [30,35]. This discrepancy could potentially be attributed to a distinct study setting. In this research, the primary reason for not using services was because "they did not think the service was needed (60.39%)", which is comparable to the study conducted in Puthyan district, where 56% revealed that they did not feel any sexual and reproductive health-related problem; hence, they did not go to the health facility [30]. Also, in KII of this study, all health workers expressed that the immature age of adolescents and the multiple changes occurring in their bodies at that age make them incapable of identifying when they need to seek services from health institutions.Another important reason, according to adolescents, was that they did not know exactly where to get services. This barrier was acknowledged by service providers too in KII, who mentioned that the adolescents did not visit the health institution because they did not have adequate information about what kinds of services are available at the health institution, and adolescents would also doubt that their problems can be addressed in local-level government health institutions. One of the most commonly reported barriers to service utilization by adolescents in the 2015 UNFPA study in Nepal was their lack of knowledge and information about AFHS [36].

From an adolescent's point of view, other reasons for not utilizing ASRH services were being shy to ask for services, fear of parents, and the opposite gender of the service provider. In KII, service providers also mentioned that at the health institutions, the sex of the service provider would determine how comfortable the adolescents were to share the problems and could also avoid coming to a health facility if the service provider were of the opposite gender. Similarly, around 43.64% of the participants who used the services were unsatisfied with the level of privacy maintained, which is also realized by service providers in KII as an important concern because of which, adolescents would rather go to private institutions or institutions that were far from their locality than be seen going to local health institutions by the community people. Specially, the research on sexual and reproductive health services: utilization pattern of adolescents in Nepal revealed that female adolescents felt embarrassed to discuss their sexual health issues with male health workers, a sentiment that male health workers also shared which was applicable in case of male adolescent and female health workers [30]. Likewise, limited utilization of ASRH services among adolescents was attributed to adolescent's fear of parents, unawareness about the benefits of the services, and lack of information [4]. Adolescents faced barriers to utilizing SRH services due to factors such as poor sexual and reproductive health knowledge, lack of privacy, experiences of shame, and attitudes from healthcare providers [37]. Studies in Nigeria and India showed that the majority of adolescents who contracted STI or other reproductive health problems did not use reproductive health services for several reasons, including shame and embarrassment, negative provider attitudes, perceived lack of confidentiality, fear of being ridiculed, and discrimination they face by some health service staff in accessing contraceptive, sexually transmitted infection services, or other services [38].

Additionally, the barriers identified in the delivery of services to adolescents were the unavailability of adolescent-friendly infrastructure such as separate rooms or proper waiting spaces, the unavailability of ASRH-trained health workers, and inadequate availability of necessary supplies. A similar finding was found in Kaski district of Nepal, where the poor physical infrastructure of health institutions was expressed as one of the barriers by all health workers while providing services, and the majority of health workers also expressed their dissatisfaction with not receiving appropriate training [39]. In a qualitative study conducted in Dhading district in Nepal, among adolescents and health care providers, the shortage of ASRH-trained health care providers in health facilities, which resulted in the allocation of non-ASRH-trained HCPs, was identified as an important factor compromising the quality of health care provided to adolescents [29]. Poor sexual and reproductive health knowledge, the lack of youth-friendly services, the lack of confidentiality of services, experiences of shame, and healthcare provider attitudes also have an influence on SRH service utilization by adolescents [37]. A study conducted in Colombia in 2015 to evaluate youth-friendly health services concluded lack of trained health workers in sexual and reproductive health and the high instability of health workers have a negative impact on the provision of services [40]. Another obstacle was the health institution's incapacity to offer comprehensive services, as they could only provide a limited range of necessary services. A review article published in 2008 reported that current sexual and reproductive health services in Nepal do not cover a whole range of SRH services that adolescents need and they need, to refer to the higher level health facility [4]. A study involving 338 adolescents from higher secondary school in Bhaktapur revealed that 30% of the participants believed the current services were insufficient to fulfill their SRH needs [21].

While observing the health institutions, there was no supply of Information, Education, Communication (IEC) materials specifically designed for ASRH; none of them had displayed the AFS logo; reporting in AFs format was null; and involvement of the adolescent in Health Facility Operation and Management Committee (HFOMC) was not in practice. Despite the health institutions' previous certification as AF institutions, their inability to maintain this status stemmed from inadequate monitoring and follow-up measures. The condition at four health institutions in Kaski was similar [39]. National Adolescent Sexual and Reproductive Health Programme; Mid-Term Evaluation Report, 2013, mentioned that the 'AFS logo signboard' is helpful to attract adolescents as it informs young people that services are available [41].

There were some missing data in the study, but the total sample size of 416 included the 10% non-response rate of the actual sample size of 378. The study was conducted after COVID-19. The schools already resumed all of the physical classes normally as in the pre-COVID-19 period but with mandatory use of masks. Therefore, this study did not have any impact of COVID-19 restrictions. However, the effects of COVID-19 on the utilization of health services by adolescents were not measured, which was the limitation of this study.

## Conclusion

Most of the participants were 15–17 years old, female, and studying in class 12. Very few were married or in a relationship. Approximately one-third of the participants engaged in discussion with their parents about various aspects of ASRH. Many knew about any health institutions that provided ASRH services in their area. For the majority of respondents, the nearest health institution was less than 30 minutes walking distance from their home. Only a few had heard about AFHS. Although the majority felt the need for ASRH services, they were still unsure about using them in the future.

The utilization of ASRH services among participants was very low. The poor utilization of SRH services was due to the inability of adolescents to identify their sexual and reproductive health needs. Some of the reasons were being uninterested in using such services, being ashamed to ask for services, and the poor availability of AF health institutions and services. This study clearly revealed that adolescents lack sufficient knowledge and a positive attitude towards using ASRH services. Therefore, to enhance their understanding and use of ASRH services, we must prioritize providing them with an adequate access to information and health services. Also, establishing adolescent-friendly infrastructure and environments along with the availability of trained health personnel to provide those services can guarantee the optimum utilization of ASRH services as per the need.

## Supporting information

**S1 Data. Data file.**
(SAV)

**S2 Data. Checklist.**
(DOCX)

## Acknowledgments

Immense gratitude towards the health coordinator of Godawari Municipality, Mr. Rewati Raj Karki, for his support to conduct the research among the health institutions of the municipality as well as all the schools for their support. The support from all the members of the faculty of public health at Manmohan Memorial Institute of Health Sciences is also appreciative. Above all, cordial thanks to all the respondents for their time, without whom this study would not have been possible.

## Author contributions

**Conceptualization:** Laxmi Gautam, Aastha Maharjan.

**Data curation:** Aastha Maharjan.

**Formal analysis:** Laxmi Gautam, Aastha Maharjan, Sujan Gautam.

**Investigation:** Aastha Maharjan.

**Methodology:** Laxmi Gautam, Aastha Maharjan, Sujan Gautam.

**Project administration:** Aastha Maharjan.

**Resources:** Laxmi Gautam, Aastha Maharjan.

**Software:** Aastha Maharjan, Sujan Gautam.

**Supervision:** Laxmi Gautam, Sujan Gautam.

**Validation:** Laxmi Gautam, Harikrishna Bhattarai, Sujan Gautam.

**Visualization:** Sujan Gautam.

**Writing – original draft:** Laxmi Gautam, Aastha Maharjan.

**Writing – review & editing:** Laxmi Gautam, Harikrishna Bhattarai, Sujan Gautam.

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
