## [Decision Letter · Decision Letter 0]

PGPH-D-24-00995

Availability and Utilization of Sexual and Reproductive Health Services among Adolescents of Godawari Municipality, Nepal: a Cross sectional Study

Dear Dr. Gautam,

Thank you for submitting your manuscript to PLOS Global Public Health. After careful consideration, we feel that it has merit but does not fully meet PLOS Global Public Health’s publication criteria as it currently stands. Therefore, we invite you to submit a revised version of the manuscript that addresses the points raised during the review process.

We look forward to receiving your revised manuscript.

Kind regards,

Adriana Biney

Academic Editor

Journal Requirements:

Additional Editor Comments (if provided):

Reviewers' comments:

Reviewer's Responses to Questions

**Comments to the Author**

1. Does this manuscript meet PLOS Global Public Health’s publication criteria?

Reviewer #1: Partly

Reviewer #2: Partly

2. Has the statistical analysis been performed appropriately and rigorously?

Reviewer #1: No

Reviewer #2: Yes

3. Have the authors made all data underlying the findings in their manuscript fully available (please refer to the Data Availability Statement at the start of the manuscript PDF file)?

Reviewer #1: No

Reviewer #2: No

4. Is the manuscript presented in an intelligible fashion and written in standard English?

Reviewer #1: Yes

Reviewer #2: No

Reviewer #1: 1. Keep keywords in alphabetical order in abstract.

2. Is your study method descriptive cross sectional? Can the inferential statistical analysis be done in descriptive studies?

3. Elaborate the sample size calculation in detail. How did you calculate the value of n as 416?

4. Sample size has been mentioned different through the study. The calculated was 416 but in table 1 variable age n is 413, n in level of education, n in relationship status is 412. In table 2 too, n is different for each variable.( perceived need of separate service-415 distance to health education-322, necessity of convenient opening hours of institution-414, Perceived Community support-415, Perceived Effect of sex of service provider-413. Why is the difference?

5. In Parents support in use of services (n=413) but n in Conversation with parents about ASRH is 416. Why is the difference? Is there error in data collection or during data management?

6. There are errors in giving asterisk (*) in table 1 for association p value. Check properly and footnote might be necessary for the meaning of asterisk (*).

7. In line number 162, sentence is not clear. The meaning of until now is not clear. Re-structure the sentence.

8. In Table 4, mention the meaning of * (multiple response).

9. Check Table 5 properly for clearly view of contents. Bullet might not be necessary.

10. Conclusion in the manuscript has been presented as summary of Results. Needs revision in conclusion part.

Reviewer #2: This is an impressive study, and the authors wanted to highlight the availability and utilization of reproductive health services among youths of classes 11 and 12. The methodology needs further clarification and justification, such as why government schools, not private schools, and why two months only. It is expected that seasonal variations will be a major factor in the accessibility and utilization of ASRH. What is the likelihood of these youths using the nearby health facilities in the municipality near the capital city. Language needs to be revised and corrected. The research did not show any impact of COVID and lockdown on access to and use of ASRH from the user and provider side

**Do you want your identity to be public for this peer review?** For information about this choice, including consent withdrawal, please see our Privacy Policy

Reviewer #1: No

Reviewer #2: No

---

## [Decision Letter · Decision Letter 1]

PGPH-D-24-00995R1

Availability and Utilization of Sexual and Reproductive Health Services among Adolescents of Godawari Municipality, Nepal: a Cross sectional Study

Dear Dr. Gautam,

Thank you for submitting your manuscript to PLOS Global Public Health. After careful consideration, we feel that it has merit but does not fully meet PLOS Global Public Health’s publication criteria as it currently stands. Therefore, we invite you to submit a revised version of the manuscript that addresses the points raised during the review process.

Please respond to the reviewer comments below. They raise concerns about the sample size. Please note it is a requirement of publication with PLOS that sample sizes are large enough to produce robust results (https://journals.plos.org/globalpublichealth/s/criteria-for-publication#loc-3).

We look forward to receiving your revised manuscript.

Kind regards,

Joanna Tindall, PhD

Staff Editor

Journal Requirements:

Additional Editor Comments (if provided):

Reviewers' comments:

Reviewer's Responses to Questions

**Comments to the Author**

Reviewer #2: (No Response)

publication criteria?

Reviewer #2: Partly

3. Has the statistical analysis been performed appropriately and rigorously?

Reviewer #2: Yes

4. Have the authors made all data underlying the findings in their manuscript fully available (please refer to the Data Availability Statement at the start of the manuscript PDF file)?

Reviewer #2: Yes

5. Is the manuscript presented in an intelligible fashion and written in standard English?

Reviewer #2: No

Reviewer #2: Study is only one directional and it is not unique to many studies conducted or published in the peer reviewed article.

In my view, exclusion of the private school students may create potential bias while the study acknowledged that private school student often has better access to resources. This is a limitation that it only reflects the one subset of the adolescent of Nepal represented from well to do societies and high income group , if i am not wrong.

The other weakness is the qualitative sample size in the study. It is quite limited (only three) and it is purposive selection of health care provider near the school, which shall be skewed towards more accessible or better equipped health facilities.

I think, it would have been more informative if the research has taken the qualitative information from the student as well to see from their perspective as well.

This study showed a correlation between the parental discussion and likelihood of using the services but fail to mention the reason behind it. The author tried to compare the parental communications findings with Ethiopia but didn’t mention why these countries would be appropriate to compare given their different societal norms and cultural differences. I think, poverty should not only be the case for comparisons.

The reader might be more interested to see beyond the basic explanations like stigma and lack of information’s (13.22%). But failed to mention the structural issues such as health system, lack of human resources etc. that country like Nepal usually suffers from.

I think, if this information were captured, we could have more information about the barriers to utilizations of ASRH. In addition, failing to measure the post-pandemic impact on SRH service utilization is a missed opportunity

**Do you want your identity to be public for this peer review?** For information about this choice, including consent withdrawal, please see our Privacy Policy

Reviewer #2: No

---

## [Decision Letter · Decision Letter 2]

PGPH-D-24-00995R2

Availability and Utilization of Sexual and Reproductive Health Services among Adolescents of Godawari Municipality, Nepal: a Cross sectional Study

Dear Dr. Gautam,

Thank you for submitting your manuscript to PLOS Global Public Health. After careful consideration, we feel that it has merit but does not fully meet PLOS Global Public Health’s publication criteria as it currently stands. Therefore, we invite you to submit a revised version of the manuscript that addresses the points raised during the review process.

The manuscript has been assessed by the two previous reviewers, and Reviewer 1 has one remaining comment.

Could you please revise the manuscript to carefully address the concerns raised?

We look forward to receiving your revised manuscript.

Kind regards,

Helen Howard

Staff Editor

Journal Requirements:

Additional Editor Comments (if provided):

Reviewers' comments:

Reviewer's Responses to Questions

**Comments to the Author**

Reviewer #1: All comments have been addressed

Reviewer #2: All comments have been addressed

publication criteria?

Reviewer #1: Yes

Reviewer #2: Partly

3. Has the statistical analysis been performed appropriately and rigorously?

Reviewer #1: Yes

Reviewer #2: Yes

4. Have the authors made all data underlying the findings in their manuscript fully available (please refer to the Data Availability Statement at the start of the manuscript PDF file)?

Reviewer #1: Yes

Reviewer #2: Yes

5. Is the manuscript presented in an intelligible fashion and written in standard English?

Reviewer #1: Yes

Reviewer #2: No

Reviewer #1: All comments have been addressed and modified. But however I have few still a question.

1. Having so many missing data, how did you ensure the validity of your results?

Reviewer #2: (No Response)

**Do you want your identity to be public for this peer review?** For information about this choice, including consent withdrawal, please see our Privacy Policy

Reviewer #1: **Yes: ** Bishwas Acharya

Reviewer #2: No

---

## [Decision Letter · Decision Letter 3]

PGPH-D-24-00995R3

Availability and Utilization of Sexual and Reproductive Health Services among Adolescents of Godawari Municipality, Nepal: a Cross sectional Study

Dear Dr. Gautam,

Thank you for submitting your manuscript to PLOS Global Public Health. After careful consideration, we feel that it has merit but does not fully meet PLOS Global Public Health’s publication criteria as it currently stands. Therefore, we invite you to submit a revised version of the manuscript that addresses the points raised during the review process.

Please note that Reviewer #1 still has outstanding concerns about your revisions in response to their requests for clarification in the methods and results; please attend to them carefully in a final revision.

We look forward to receiving your revised manuscript.

Kind regards,

Avanti Dey, PhD

Staff Editor

Journal Requirements:

Additional Editor Comments (if provided):

Reviewers' comments:

Reviewer's Responses to Questions

**Comments to the Author**

Reviewer #1: (No Response)

publication criteria?

Reviewer #1: Yes

3. Has the statistical analysis been performed appropriately and rigorously?

Reviewer #1: Yes

4. Have the authors made all data underlying the findings in their manuscript fully available (please refer to the Data Availability Statement at the start of the manuscript PDF file)?

Reviewer #1: Yes

5. Is the manuscript presented in an intelligible fashion and written in standard English?

Reviewer #1: Yes

Reviewer #1: I am not satisfied with your answer. As per you, "The total 416 samples included the 378 actual sample sizes and added 10% of the sample size.""There are some missing data in the study, but the total sample size of 416 included the 10%

non-response rate of the actual sample size of 378, so there should be no problem with the

validity of the study."

1. Do you mean missing data and non-respondents are same? Can the data which has been missed be referred as non respondents?

2. Next the data missed may be of different respondents for different variable. Can you conclude saying missing data as non respondents on that case? You need to consult a statistician before giving any judgement as this manuscript is for the publication in a renowned journal.

3. In Table 2: Need and External Environmental factors of ASRHS, the sample size of variable 'utilization

Non Distance to health institution' is 322 which is much below 378. How can you ensure the validity on that case?

**Do you want your identity to be public for this peer review?** For information about this choice, including consent withdrawal, please see our Privacy Policy

Reviewer #1: No

---

## [Decision Letter · Decision Letter 4]

Availability and Utilization of Sexual and Reproductive Health Services among Adolescents of Godawari Municipality, Nepal: a Cross sectional Study

PGPH-D-24-00995R4

Dear Ms Gautam,

We are pleased to inform you that your manuscript 'Availability and Utilization of Sexual and Reproductive Health Services among Adolescents of Godawari Municipality, Nepal: a Cross sectional Study' has been provisionally accepted for publication in PLOS Global Public Health.

Best regards,

Julia Robinson

Executive Editor

Reviewer Comments (if any, and for reference):

Reviewer's Responses to Questions

**Comments to the Author**

Reviewer #1: All comments have been addressed

publication criteria?

Reviewer #1: Yes

3. Has the statistical analysis been performed appropriately and rigorously?

Reviewer #1: Yes

4. Have the authors made all data underlying the findings in their manuscript fully available (please refer to the Data Availability Statement at the start of the manuscript PDF file)?

Reviewer #1: Yes

5. Is the manuscript presented in an intelligible fashion and written in standard English?

Reviewer #1: Yes

Reviewer #1: I have no comments.

**Do you want your identity to be public for this peer review?** For information about this choice, including consent withdrawal, please see our Privacy Policy

Reviewer #1: No
